# Thioredoxin Reductase and Organometallic Complexes: A Pivotal System to Tackle Multidrug Resistant Tumors?

**DOI:** 10.3390/cancers15184448

**Published:** 2023-09-06

**Authors:** Michèle Salmain, Marie Gaschard, Milad Baroud, Elise Lepeltier, Gérard Jaouen, Catherine Passirani, Anne Vessières

**Affiliations:** 1Sorbonne Université, CNRS, Institut Parisien de Chimie Moléculaire (IPCM), 4 Place Jussieu, F-75005 Paris, France; michele.salmain@sorbonne-universite.fr (M.S.); marie.gaschard_stefanelli@sorbonne-universite.fr (M.G.); gerard.jaouen@sorbonne-universite.fr (G.J.); anne.vessieres@sorbonne-universite.fr (A.V.); 2Micro & Nanomedecines Translationnelles (MINT), University of Angers, Inserm, The National Center for Scientific Research (CNRS), SFR ICAT, F-49000 Angers, France; m.baroud121@gmail.com (M.B.); elise.lepeltier@univ-angers.fr (E.L.)

**Keywords:** organometallic complexes, *N*-heterocyclic carbene, auranofin, gold, ferrocene, thioredoxin reductase, cancer

## Abstract

**Simple Summary:**

The identification of biological targets is an essential step in deciphering the mechanism of action of anticancer drugs. In this review, we chose to study the relationship between the inhibition of thioredoxin reductase (TrxR), a key enzyme in maintaining the redox balance of cells, and the cytotoxic effects of two groups of organometallic complexes. The first group is essentially composed of Au(I) and Au(III) complexes and the second one comprises metallocifens (organometallic complexes derived from tamoxifen). The results show that these two groups interact differently with TrxR at the molecular level. Even if the contribution of TrxR inhibition to the cytotoxicity of complexes is clearly established for many of them, the number of complexes for which TrxR inhibition plays a predominant role appears quite limited. Eventually, the antiproliferative activity of most of the complexes appears to stem from the interaction with several targets, a favorable strategy to tackle MDR tumors.

**Abstract:**

Cancers classified as multidrug-resistant (MDR) are a family of diseases with poor prognosis despite access to increasingly sophisticated treatments. Several mechanisms explain these resistances involving both tumor cells and their microenvironment. It is now recognized that a multi-targeting approach offers a promising strategy to treat these MDR tumors. Inhibition of thioredoxin reductase (TrxR), a key enzyme in maintaining redox balance in cells, is a well-identified target for this approach. Auranofin was the first inorganic gold complex to be described as a powerful inhibitor of TrxR. In this review, we will first recall the main results obtained with this metallodrug. Then, we will focus on organometallic complexes reported as TrxR inhibitors. These include gold(I), gold(III) complexes and metallocifens, i.e., organometallic complexes of Fe and Os derived from tamoxifen. In these families of complexes, similarities and differences in the molecular mechanisms of TrxR inhibition will be highlighted. Finally, the possible relationship between TrxR inhibition and cytotoxicity will be discussed and put into perspective with their mode of action.

## 1. Introduction

Despite significant improvements in the management of many cancers with the advent of targeted therapies and immunotherapy, the emergence of multidrug resistance (MDR) is accountable for over 90% of the first-line therapeutic failures and currently represents one of the greatest challenges in the field [1,2,3,4]. Numerous mechanisms have been ascertained as responsible for MDR in cancers, such as DNA repair capacity, growth factor overproduction, xenobiotics metabolism, enhanced drug efflux and genetic factors [4,5,6,7,8]. MDR cancers include glioblastoma [9], melanoma [10], non-small-cell lung cancer (NSLC) [11], and triple negative breast cancer (TNBC) [12]. To circumvent resistance phenomena and side effects associated with clinically available chemotherapeutic agents, the current trend is to move from a single-targeted to a multi-targeted therapeutic approach [13], which will harm not only the tumor but also its microenvironment [14].

The first chemotherapy agents were designed to stop DNA replication, resulting in cell proliferation arrest. The most representative examples in this field are the platinum complexes (cisplatin and its two related molecules, carboplatin and oxaliplatin), which still form the first-line treatment of various cancers. Indeed, the intrastrand crosslinking of guanine residues by platinum coordination causes bending of the DNA strand, replication arrest and eventually apoptosis [15,16]. The effectiveness of platinum compounds stimulated the design of numerous metal complexes, both inorganic and organometallic (molecules with direct metal–carbon bonds), with the aim to decrease side effects and alleviate intrinsic and acquired resistance mechanisms [16,17,18]. Complexes of interest are generally selected according to their strong antiproliferative effects on various cell lines, including MDR ones. It soon became apparent that their effects are unrelated to DNA but mediated by interactions with proteins such as receptors or enzymes [13,19,20]. Thioredoxin reductase is one enzyme among others (cathepsin B; glutathione S-transferase, kinases) that is targeted by metal complexes. TrxR is a ubiquitous enzyme that, combined with thioredoxin (Trx), plays an essential role in maintaining the redox balance within cells, by a mechanism that will be developed in detail below [21]. It should also be noted that overexpression of TrxR in cancer cells has been extensively reported in the literature [22,23,24]. The quantification of TrxR in tumors showed that its level of expression is variable and cancer-dependent. Its high level in tumors (lung, breast and liver) is generally associated with a poor prognosis in terms of patient survival [25,26,27]. Accordingly, the level of TrxR in mouse serum is higher in mice with hepatocellular carcinoma (HCC) with respect to healthy mice, opening the possibility of its use as a tumor marker [22].

The search for inhibitors of the thioredoxin/thioredoxin reductase system as anti-cancer agents seems therefore amply justified [28,29,30]. Moreover, the possibility to reverse MDR by modulating ROS production via TrxR inhibition has been underlined [7]. TrxR inhibitors described in the literature have extremely diverse chemical structures and are the subject of various reviews [23,31,32,33]. They include arsenic trioxide, organic molecules, natural products (curcumin) [34,35], as well as a large number of metal complexes, both inorganic and organometallic [36].

We choose to focus this review on the contribution of organometallic complexes to this field and on the putative relationship between TrxR inhibition and cytotoxicity. To the best of our knowledge, this topic has never been approached from this angle. Organometallic inhibitors of TrxR belong to two families of molecules that interact either by direct metal coordination to TrxR or as Michael acceptors. In the first family, complexes of gold(I) and gold(III) are the most represented [37,38,39], followed by complexes of other metals (Ag, Ru, Ir, Os). The second family comprises the metallocifens mainly represented by ferrocifens [23]. Before reviewing these inhibitors and discussing their mechanism of action, we will begin with a historical review of the discovery of TrxR, the identification of its molecular structure and the mechanism of its biosynthesis.

## 2. Thioredoxin Reductase: Preparation, Characterization and Biological Properties

### 2.1. Purification and Biosynthesis of TrxR

The enzymatic activity of the flavoenzyme TrxR and its substrate thioredoxin (Trx) were both discovered in *E. coli* in 1964 [40]. The first mammalian TrxR was isolated and characterized from rat liver in 1982 [41], whereas the human enzyme was isolated from human placenta in 1993 [42]. A more expedite purification of hTrxR1 from human placenta was reported in 1998 [43]. The sequence of hTrxR1 was published in 1995 [44]. In 1996, a selenoenzyme isolated from lung carcinoma cells was identified as TrxR1 [45]. The biologists Holmgren and Arnér played a major role in the development of this research. Rigobello and Bindoli also markedly contributed to the area by isolating and characterizing the mitochondrial isoform of TrxR [46], then by making a bridge between biological studies and gold-based inhibitors, starting from auranofin (see Section 3) [47,48].

TrxR is a member of the twenty-five human selenoproteins. These proteins possess the unique particularity of containing at least one selenocysteine (Sec or U), which is considered as the 21st amino acid [49,50,51]. The mechanism of biosynthesis of selenoproteins, and more specifically the insertion of Sec into nascent proteins, was first elucidated in *E. coli* by Böck in 1991 [52], and then in eukaryotic cells by Hatfield and Söll in 2006 [53,54]. Despite its name, Sec is derived from serine (Ser or S), although being structurally close to cysteine (Cys or C). The mechanism by which selenocysteine is inserted into a polypeptide chain is summarized in Figure 1 [55]. Two steps are crucial: first, the integration of selenium onto the tRNA[Ser][Sec], and second, the incorporation of a selenocysteine coded by the opal codon UGA, which normally codes for the termination of the translation process. This is due to the presence of the selenocysteine insertion sequence (SECISBP2) on the stem loop at the 3′ extremity of the mRNA [55]. Several reviews report in more detail the mechanism of biosynthesis of selenoproteins [30,49,50,56].

**Figure 1 cancers-15-04448-f001:**
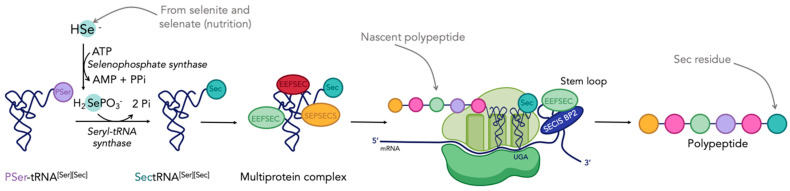
Mechanism of selenoproteins biosynthesis in eukaryotic cells (adapted from [55]). Figure created with BioRender.com.

According to this mechanism, selenium appears as an essential trace element determining the presence of selenocysteine in cells, and by extension the presence of all the selenoproteins. Then, it is important to pay attention to the selenium concentration in the cell culture medium when looking at the activity of selenoproteins (such as TrxR or glutathione peroxidase, GPx) [57]. Few studies have been carried out on the correlation between the quantity of Se and their activity, and in turn, on drugs’ cytotoxicity [58,59,60]. As an example, in 2012, Arnér and collaborators brought to light the subtle effect of selenium concentration on the antiproliferative activity of cisplatin. In fact, they demonstrated that selenium supplementation sensitized cells to cisplatin [59], and the effect of selenium was dose- and cell-dependent. Consequently, this parameter needs to be considered when working with selenoproteins.

### 2.2. Structure of the Isoforms 1 and 2

Mammalian TrxR exists in two main isoforms, a cytosolic form (TrxR1) and a mitochondrial form (TrxR2). A third isoform, TrxR3 or TGR, is only present in testis tissue and displays an additional glutathione reductase (GR) function. The monomer of the human cytosolic isoform TrxR1 is a 499-amino acid polypeptide, while the monomer of the mitochondrial isoform TrxR2 is a 532-amino acid protein with an 84% homology to TrxR1. This latter enzyme comprises an additional 33-amino acid sequence at the N-terminus responsible for its addressing to mitochondria [61]. Like TrxR1, TrxR2 catalyzes the reduction of Trx2, the mitochondrial isoform of Trx1. Isoforms 1 and 2 of TrxR assemble into homodimers of ca. 2 × 55 kDa. Each subunit includes two redox active catalytic sites, a dithiol/disulfide motif present in the -CVNVCG- hexapeptide sequence located in the N-terminal domain and a selenol-thiol/selenenylsulfide motif formed by the vicinal cysteine and selenocysteine residues of the C-terminal tetrapeptide sequence GCUG. This selenocysteine is only present in the TrxR of higher organisms, which makes it an essential feature of this enzyme. From a chemical point of view, the lower pK_a_ of the selenol/selenolate couple (=5.2) with respect to the thiol/thiolate couple (=8.5) confers to selenocysteine a unique reactivity responsible for the biological activity of TrxR. TrxR also includes one strongly bound FAD and one NADPH binding site per subunit in addition to the two redox centers (Figure 2a). This arrangement facilitates the general flow of reducing entities from NADPH to a wide range of substrates occurring during catalysis [31].

**Figure 2 cancers-15-04448-f002:**
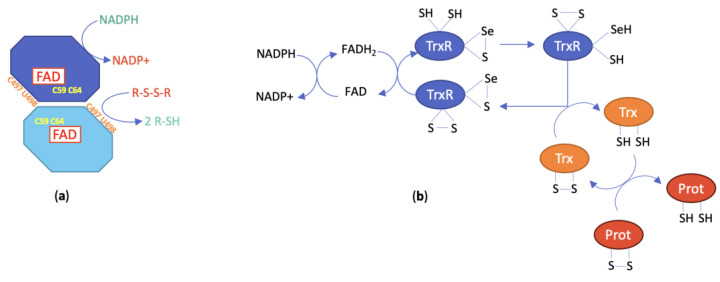
(**a**) Schematic representation of the structure of TrxR; (**b**) catalytic mechanism of TrxR (numbering is taken from hTrxR1 and hTrx1).

The first crystal structure of rat TrxR1 (U498C mutant) was published in 2001 [62]. The crystal structure of human TrxR1 (U498C mutant) was published by Becker in 2007 [63] and eventually the crystal structure of wild-type human TrxR1 was reported by Arnér in 2009 [64]. In all cases, the two subunits of TrxR1 assemble in a head-to-tail arrangement with each C-terminal redox center positioned at the interface between the two monomers (Figure 3). The topology and the active site of the rat form was shown to be close to that of glutathione reductase (GR, see Section 3.2), another pyridine nucleotide disulfide oxidoreductase. However, GR noticeably lacks the second redox center. The structure of the human form of TrxR also revealed that the intrinsic flexibility of C-terminal peptide sequence was essential to the catalytic mechanism.

**Figure 3 cancers-15-04448-f003:**
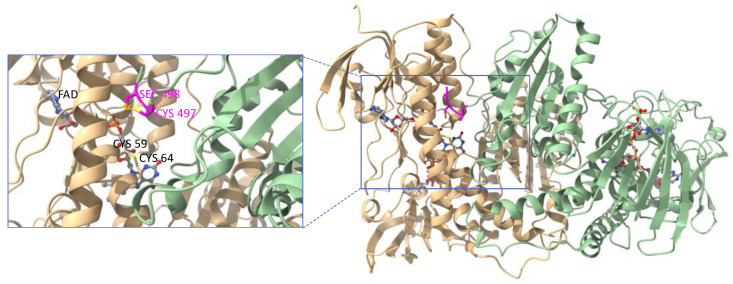
Ribbon representation of the crystal structure of the oxidized form of wild-type human TrxR1 (PDB file: 3EAO). Subunit A is shown in light green and subunit B in light beige. Each subunit binds one molecule of FAD and one molecule of NADPH represented as ball-and-stick models. Close-up view of one of the C-terminal redox centers (tetrapeptide sequence colored in magenta) positioned at the interface between the subunits. Figure generated with ChimeraX [65].

### 2.3. Catalytic Mechanism of TrxR

The catalytic mechanism of TrxR has been established from structural analyses and biochemical studies. It can be roughly represented by the scheme in Figure 2b. More extensive information on the detailed model can be found in [34]. For the catalysis to take place, the homodimeric structure is compulsory as the dithiol pair of the N-terminal redox center reduces the selenenylsulfide redox center of the opposite subunit to generate the selenol-thiol reducing pair. In turn, this pair will reduce the active site disulfide of Trx as well as other substrates [31]. The reaction of electrophilic agents will preferentially occur with the Sec residue of the NADPH-reduced TrxR form, leading to the accumulation of oxidized Trx and the deregulation of cellular redox homeostasis. Indeed, both the thioredoxin and the glutathione systems are responsible for cell redox balance [30]. The inhibition of TrxR translates into the overproduction of ROS that can be partly ascribed to the conversion of TrxR into a pro-oxidant enzyme by acquisition of the NADPH oxidase property [66]. The Trx/TrxR system modulates the activity of several central transcription factors involved in cell proliferation and cell death, including NF-kB, p53 and Nrf2 [31]. Consequently, TrxR inhibition will affect the function of both healthy and tumoral cells and is likely to induce numerous in vivo effects.

## 3. Representative Inorganic Inhibitors of TrxR

### 3.1. Auranofin

Auranofin (1-thio-β-D-glucopyranosatotriethylphosphine gold-2,3,4,6-tetraacetate, **AuF**, commercial name Ridauta^®^, [Fig cancers-15-04448-ch001]) was initially introduced in the clinic as an orally-administered drug for the treatment of rhumatoid arthritis [67]. Since then, **AuF** has undergone a number of repurposing studies for other therapeutic indications [68], including cancer [69].

Auranofin displays broad-range anticancer properties both in cellulo and in vivo [70]. Early in vivo studies showed that the intraperitoneal administration of auranofin to mice inoculated with P388 lymphocytic leukemia cells increased their life span by 59% at the optimal dose of 12 mg/kg. Auranofin also displays strong cytotoxic activity on cells, with IC_50_ = 1.01 and 0.43 µM, respectively, in cisplatin-sensitive and -resistant cancer cells [71,72,73,74]. This anticancer property has further been linked to the inhibition of both isoforms of TrxR, leading to the overproduction of ROS and the disruption of cellular redox homeostasis [75]. Of note, TrxR acts as a soft base owing the presence of thiols and selenol, sulfur and selenium having high affinity for soft acids including Au(I) complexes [76].

The inhibition of TrxR by **AuF** was first highlighted in 1998 by Becker when the enzyme was purified from human placenta, with IC_50_ in the nanomolar range (IC_50,hTrxR_ = 20 nM and IC_50,TrxR_ = 16.7 nM in A2780 cells) [43,73]. **AuF** is much less active on GR and GPx since the IC_50′_s are in the micromolar range (respectively, IC_50,hGR_ = 40 µM and IC_50,hGPx_ > 100 µM) [43]. It was also noticed that **AuF** was able to inhibit TrxR only in its reduced from, i.e., after treatment with NADPH [77]. Furthermore, **AuF** is equally active on TrxR1 and TrxR2 [78]. Hence, **AuF** was suspected to primarily interact with the selenocysteine of the C-terminal redox active site. Indeed, mass spectrometric analysis of the reaction product of **AuF** with the C-terminal tetrapeptide of TrxR demonstrated that up to two AuPEt_3_ motifs were bound to the peptide, most likely by coordination of the S and the Se atoms of Cys and Sec [34]. MALDI-TOF MS analysis of the product of reaction of **AuF** and TrxR1 (at 10:1 ratio) showed that approximately three AuPEt_3_ bound to the enzyme [21], indicating that U498 is not the sole amino acid targeted by **AuF**.

In the last four years, several proteomic studies were carried out to uncover the cellular targets of **AuF** and delineate its mechanism of action. Expression proteomics revealed that the oxidative stress marker HMOX became upregulated upon incubation of HT116 cells with **AuF** (1 µM). This finding, together with induced mitochondrial dysfunction, provided confirmation that TrxR was indeed the main target of **AuF** [79]. A Combination of orthogonal chemical proteomics approaches, namely, FITExP (Functional Identification of Target by Expression Proteomics), multiplex redox proteomics and thermal proteome profiling, confirmed again that TrxR1 belongs to the main **AuF** targets together with NFKB2 and CHORDC1, the first one being responsible for the anti-inflammatory effect of **AuF** [80]. Additional chemical proteomics studies revealed that other protein targets than TrxR may be responsible for cell death induced by **AuF** [81]. Finally, according to a cysteine redox proteomics study, TrxR1 was found to be upregulated in cells exposed to **AuF**, consistent with its inhibition by **AuF,** while the level of TrxR2 remained unchanged [82].

The discovery that THE inhibition of TrxR by auranofin is responsible for its antitumor activity triggered the development of a wide range of cytotoxic gold(I) and gold(III) complexes. Several reviews covering this area have been published within the last five years [37,38,39,76,83].

### 3.2. Other Potent Inorganic Inhibitors

Many other coordination complexes have been identified as TrxR inhibitors [36]. Here, we focus on three different compounds, for which structural data are available. Becker’s group introduced the gold(I) complex **Phos-Au** ([Fig cancers-15-04448-ch001]) as a highly potent inhibitor of both human GR (IC_50,hGR_ = 1 nM) and wild-type TrxR (IC_50,hTrxR_ = 6.9 nM) [84]. Interestingly, the inhibition property of **Phos-Au** toward TrxR drastically decreased when U498 was mutated to C (IC_50,TrxR_ = 900 nM). **Phos-Au** was shown to impede the proliferation of glioblastoma cells with IC_50_ between 5 and 13 µM. The crystal structure of the adduct of GR and **Phos-Au** was solved at a good resolution (Figure 4), and two gold(I) ions happened to be bound per GR monomer. The first one bound to the two sulfur atoms of the active site cysteines 58 and 63 with decoordination of the phosphole and the chlorido ligands. This binding site occupies the same position as the N-terminal redox center of human TrxR1. The second gold(I) was bound to the surface C284 via coordination of its sulfur atom and displacement of the Cl ligand.

The inhibitory activity of the terpyridine Pt(II) complex **TP–Pt** ([Fig cancers-15-04448-ch001]) was determined on the U498C mutant of hTrxR1 (IC_50, TrxR_ = 74 nM) [85]. **TP–Pt** showed antiproliferative properties on HeLa cells with a moderate IC_50_ of 16.5 µM after 72 h. The crystal structure of the adduct of hTrxR1 (U498C mutant) and **TP–Pt** showed that a dative bond was formed between the sulfur atom of C498 and the Pt(II) center, which was confirmed by mass spectrometry analysis of the tryptic peptides. A secondary π-stacking interaction was also observed between the terpyridine ligand and the indole substituent of W114 of the neighboring subunit (Figure 5).

**Figure 4 cancers-15-04448-f004:**
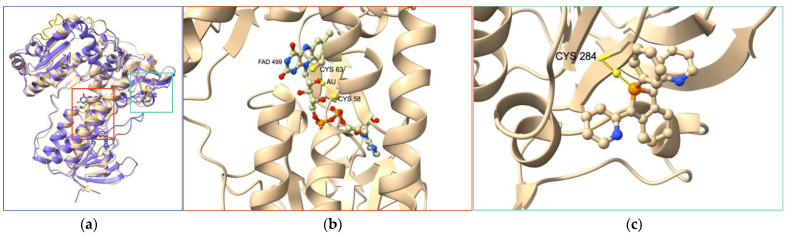
(**a**) Superimposition of crystal structures of hTrxR1 (monomer, purple, PDB file 2j3n) and hGR treated by Au complex **Phos-Au** (light beige, PDB file 2aaq). (**b**,**c**) Close-up views of the gold-binding sites in hGR. One gold atom is S-coordinated to C58 and C63 forming the redox center of hGR; this redox center occupies the same position as the N-terminal redox center in hTrxR1; the second gold atom is S-coordinated to C284 and the phosphole ligand of **Phos-Au**; the peptide sequence comprising the C-terminal redox center of hTrxR1 is colored in orange.

**Figure 5 cancers-15-04448-f005:**
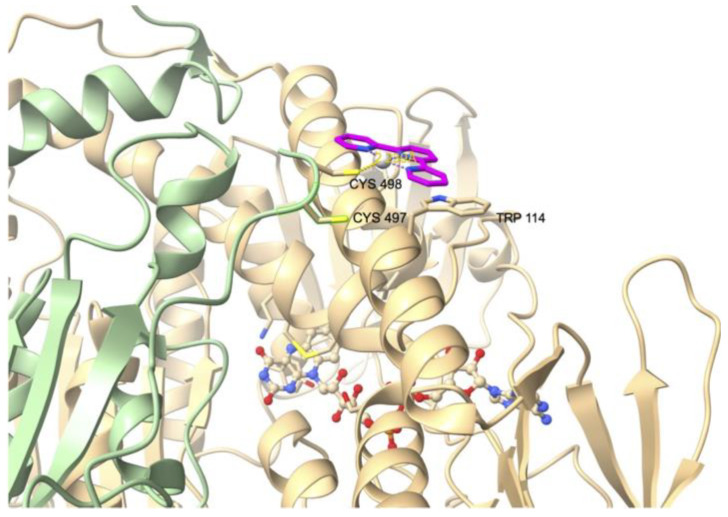
Crystal structure of the adduct of hTrxR1 (U498C mutant) with platinum complex **TP–Pt** (PDB file: 2zzb).

Very recently, a cRGD-bound, tetranuclear gold cluster was designed to selectively target tumor cells and exert antitumor activity [86]. It was found to display a strong affinity for pure hTrxR. A cell line overexpressing hTrxR carrying a Strep-tag was produced and incubated with the gold cluster. Mass spectrometric analysis of hTrxR pulled out by affinity chromatography provided evidence for the formation of a mono-gold adduct, indicating that the gold cluster was metabolized in cells. Cryo-electron microscopy of the gold-TrxR adduct revealed that the gold ion was unexpectedly bound to a surface cysteine residue. Of note, the C-terminal peptide sequence of TrxR could not be modeled owing to its conformational flexibility. The authors hypothesized that the binding of the gold ion induced a shift of the α-helix formed by residues 96–124, preventing the further binding of Trx.

Following the discovery that auranofin had antiproliferative and antitumor activity, numerous gold(I) complexes were studied [87]. Despite their great potential as anticancer agents, most of them suffer from an absence of selectivity resulting in high general toxicity. Nevertheless, among the family of delocalized lipophilic cations (DLCs) targeting the mitochondria [88], the monocationic, tetracoordinated Au(I) complex **[Au(d2pypp)_2_]Cl** (d2pypp = 1,3-bis(di-2-pyridylphosphino)propane) ([Fig cancers-15-04448-ch001]) stands out by being both highly cytotoxic toward breast cancer cells and ineffective towards normal breast cells. This complex induces apoptotic cell death via the mitochondrial pathway. Its selectivity is related to its greater ability to inhibit both Trx and TrxR in breast cancer cells [89]. This research highlights the involvement of TrxR in the mechanism of action of transition metal complexes.

## 4. Inhibition of TrxR and Cytotoxicity of a Selection of Organometallic Complexes

Among the abundance of literature, we chose to focus on a selection of 45 organometallic complexes that we consider as representative of the state of the art in the field. Our goal is to establish a possible correlation between the ability of the complexes to inhibit TrxR (in its isolated form, in cells or even in tumor) and their effects on cancer cell viability. These molecules are represented in [Fig cancers-15-04448-ch002] and [Fig cancers-15-04448-ch003]; their IC_50_ on isolated TrxR (or cellular TrxR) and their cytotoxicity are gathered in the following tables. The first group (Table 1) includes the linear NHC-gold(I) and silver(I) complexes having as second ligand L a second NHC, a thiolate or a chlorido ligand. The second group (Table 2) comprises other organometallic gold(I) complexes and gold(III) complexes. The third group (Table 3) comprises organometallic complexes of other transition metals. Finally, the fourth group consists of a selection of metallocifens (see table in Section 5).

### 4.1. Gold(I) Complexes: NHC^Au(I)^L and PPh_3_^Au(I)^Alkyne

Berners-Price established for the first time the link between cytotoxicity, mitochondrial disruption and selective cellular thioredoxin reductase inhibition [90]. They introduced the cationic (Bis)NHC gold complex **1** that not only inhibits TrxR in cells but also induces a concentration-dependent decrease in the viability of two breast cancer cells, while having no activity on normal cells. The authors hypothesized on a possible mechanism where **1** interacts with TrxR by losing successively its two NHC ligands, to form a linear Cys-Au-Sec complex (Figure 6). This mechanism is consistent with the binding of **Phos-Au** to GR described in Section 3.2.

In 2010, Ott studied some NHC^Au(I)^L complexes with different ligands (L = Cl, NHC or PPh_3_). The correlation between a low IC_50_ value for the inhibition of TrxR and a high cytotoxicity on cancer cells was not always found. Indeed, for some cases, the toxicity of the complex on cancer cells is not associated with a large inhibition of the enzyme (complex **3**, IC_50,TrxR_ = 0.36 μM; IC_50_ = 4.6 μM or complex **4**, IC_50,TrxR_ = 4.9 μM; IC_50_ = 0.8 μM, both on MCF-7 cells) [91,92]. Conversely, complexes **5** (with L = PPh_3_) [92], **2** [93] and **8** (L = NHC ligand) [94] are all efficient cytotoxic agents and TrxR inhibitors, with respectively IC_50,TrxR_ = 0.66, 1.2 or 0.7 μM and IC_50_ = 0.9, 1.0 or 0.1 μM on MCF-7 cells. Complex **5** is considered as a DLC and, in contrast to compound **3**, is preferentially accumulated in mitochondria [91]. Ott also introduced neutral PPh_3_^Au(I)^alkyne complexes. Complexes **16**–**18** show similar trends as **2**, **5**, and **8** (IC_50,TrxR_ = 45, 359 or 47 nM and IC_50_ = 1.0, 2.2 or 0.8 μM on MCF-7 cells) [95]. The authors also determined that those complexes were inactive on GR. To confirm the binding of complexes **16** and **18** to the Sec residue of TrxR, they carried out a mass spectrometry analysis with a Sec-containing peptide, and were able to conclude that the Sec residue is indeed the major gold-binding site [95]. A similar experiment was performed with compound **5** [92].

**Table 1 cancers-15-04448-t001:** Inhibition of TrxR and cytotoxicity on cells of a selection of NHC gold and silver complexes (NHC^Au(I)^L, NHC^Ag(I)^L).

Type of Complex	L	#	Entry	IC_50,TrxR_	Toxicity on Cancer Cells		References
					IC_50_	Cell Lines	
NHC^Au(I)^L	NHC	**1**	1	5 µM ^(a)^	25 µM ^(b)^	MDA-MB-231 ^(c)^	[90,96]
	NHC	**2**	2	1.2 µM	1 µM	MDA-MB-231	[93]
	Cl	**3**	3	0.36 µM	4.6 µM	MCF-7	[91,92]
	NHC	**4**	4	4.9 µM	0.8 µM	MCF-7	[91,92]
	PPh_3_	**5**	5	0.66 µM	0.9 µM	MCF-7	[91,92]
	NHC	**6**	6	2.2 µM	0.25 µM	PC3	[97]
	NHC	**7**	7	2.1 µM	0.46 µM	PC3	[97]
	NHC	**8**	8	0.7 µM	0.10 µM	MCF-7	[94]
	NHC	**9**	9	8 µM ^(b)^	0.5 µM	Ishikawa ^(d)^	[98]
	Thiolate	**10**	10	4.9 nM	3.2 µM	A2780S	[19]
			11		4.9 µM	A2780R	[19]
	Thiolate (CpTiCl_2_)	**11**	12	<5 µM ^(e)^	9.8 µM	PC-3	[99]
	Cl	**12**	13	12.6 nM	5.2 µM	A2780S	[100]
NHC^Ag^L	Cl	**13**	14	5.9 nM	3.3 µM	A2780S	[100]
	NHC	**14**	15	2.4 nM	0.09 µM	A2780cis/CP70	[101,102]
			16		0.44 µM ^(c)^	A2780	[101,102]
	NHC	**15**	17	12.5 nM	14.6 µM	MCF-7	[103]

^(a)^ on MDA-MB-231 cells; ^(b)^ estimated values; ^(c)^ after 24 h; ^(d)^ in vivo studies (cf. Section 4.6); ^(e)^ on PC-3 cells.

**Table 2 cancers-15-04448-t002:** PPh_3_^Au(I)^alkyne complexes, multi-modal complexes and Au(III) complexes.

Type of Complex	#	Entry	IC_50,TrxR_ ^(a)^	Toxicity on Cancer Cells		References
				IC_50_	Cell Lines	
PPh_3_^Au(I)^alkyne	**16**	18	45 nM	1.0 µM	MCF-7	[95]
	**17**	19	359 nM	2.2 µM	MCF-7	[95]
	**18**	20	47 nM	0.8 µM	MCF-7	[95]
	**19**	21	2.8 nM	0.03 µM	MCF-7	[104]
Multi-modal	**20**	22	2.7 µM	10 µM	A2780 ^(c)^	[105]
		23	17 µM in A2780			[105]
	**21**	24	10 nM	ca. 6 µM	HeLa ^(c)^	[106]
	**22**	25	2.5 µM ^(b)^	0.026 µM	A2780	[107]
	**23**	26	^(d)^	ca. 3 µM	HeLa	[108]
Au(III)	**24**	27	19 nM	22 µM	HepG2 ^(c)^	[109]
		28	32 µM in HepG2			[109]
	**25**	29	2.7 nM	0.21 µM	HepG2 ^(c)^	[109]
		30	7.3 µM in HepG2			[109]
	**26**	31	2 µM	0.42 µM	HeLa	[110]
	**27**	32	3 nM on TrxR1	13 µM	SKOV-3	[111]
		33	60 nM on TrxR2			[111]
	**28**	34	2 µM	0.12 µM	A2780	[112]
	**29**	35	0.21 µM TrxR2	3 µM	A2780S	[113,114]
		36		7 µM	A2780R	[113,114]
	**30**	37	0.28 µM TrxR2	1 µM	A2780S	[113,114]
		38		7 µM	A2780R	[113,114]
	**31**	39	18 nM TrxR1	47 µM	A2780	[115]
	**32**	40	9 nM TrxR1	38 µM	A2780	[115]

^(a)^ Measured in vitro except when noticed; ^(b)^ estimated values; ^(c)^ in vivo studies (cf. Section 4.6); ^(d)^ addition of siRNA anti-TrxR decreased the cytotoxicity of the complex.

**Table 3 cancers-15-04448-t003:** Complexes of other metals.

Metal	#	Entry	IC_50,TrxR_	Toxicity on Cancer Cells		References
				**IC_50_**	**Cell Lines**	
Ru	**33**	41	0.78 µM	2.07 µM	MCF-7	[116]
		42		2.2 µM	HT-29	[116]
Ru	**34**	43	285 nM	3.5 µM	HT-29	[117]
		44		1.9 µM	MCF-7	[117]
Ru	**35**	45	1.42 µM	18 µM	HT-29	[117]
		46		18.4 µM	MCF-7	[117]
Ir	**36**	47	68 nM	5.1 µM	HT-29	[117]
		48		3.4 µM	MCF-7	[117]
Ir	**37**	49	1.48 µM	19 µM	HT-29	[117]
		50		35 µM	MCF-7	[117]
Ru	**38**	51	4.1 µM	1 µM	A2780	[118]
		52		1.1 µM	A2780R	[118]
		53		1.6 µM	A2780ADR	[118]
Os	**39**	54	30% inhibition at 5 µM ^(a)^	2.4 µM	MDA-MB-231	[119]

^(a)^ in MDA-MB-231.

**Figure 6 cancers-15-04448-f006:**
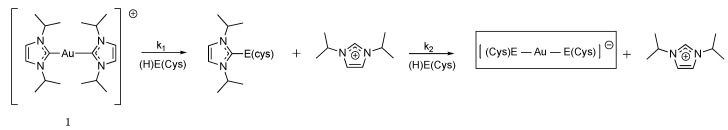
Chemical reaction between complex **1** and the amino acid cysteine or selenocysteine (E = S or Se), adapted from [90].

The involvement of the Sec residue of TrxR in the inhibition mechanism was also proven by comparing the activity on TrxR from *E. coli* devoid of Sec to the one on the rat enzyme. Complex **8** displayed an IC_50,TrxR_ of 0.7 μM on rat versus an IC_50,TrxR_ of 48 μM on *E. coli* [94]. Additionally, the BIAM assay is widely used by Rigobello and Bindoli [19]. In this test, BIAM (biotin-conjugated iodoacetamide) can selectively alkylate TrxR depending on pH. At pH 6.0, only selenocysteine and low pK_a_ cysteines can be alkylated, while at pH 8.5, both selenocysteine and cysteines can be derivatized by BIAM. Complex **19** (with diphenyl(thiophen-2-yl)phosphane ligand instead of triphenylphosphane) is one of the most efficient compounds in the list of selected molecules with IC_50,TrxR_ = 2.8 nM and IC_50_ = 0.03 μM on MCF-7 cells [104]. Of note, the replacement of a phenyl group by a thiophene group on the PPh_3_ allows an enhanced p–π conjugation, and then increases the complex stability in the presence of albumin, known to tightly bind gold complexes via its cysteine residue. Complex **19** was formulated in pH-sensitive micelle-like nanoparticles known to elicit autophagy. Such encapsulation was found to dramatically exacerbate the cytotoxic properties of **19**, inducing cell death by autophagy and apoptosis [120].

The (Bis)NHC Au(I) complexes **6** and **7** introduced by Hemmert have IC_50,TrxR_ = 2.2 and 2.1 μM and IC_50_ = 0.25 and 0.46 μM, respectively, on PC-3 and HepG2 cells [97,121]. Consistently with compound **1**, their ROS-dependent toxicity is mostly observed on cancer cells and less pronounced on normal cells, and is partially abolished by the pre-incubation of *N*-acetylcysteine (NAC). The authors also noticed a positive relationship between lipophilicity and cytotoxicity, at the expense of selectivity [121]. Let us notice that complex **7** is equally active on isolated TrxR and TrxR in HepG2 cells [97]. The heteroleptic Bis(NHC) Au(I) complex **9** showed a dose-dependent inhibition of TrxR in Ishikawa endometrial cancer cells, and ROS production after 8 h of treatment. The authors also noticed a decrease in the expression of both Nrf2 and its downstream member TrxR [98].

NHC gold(I) complexes comprising a thiolate ligand have also been reported (complexes **10** and **11**) [19,99]. Complex **10** showed an extremely strong inhibition of TrxR1 (IC_50,TrxR_ = 4.9 nM), as well as TrxR2 and GR, albeit to a much lesser extent. This latter finding could prove its interaction with the Sec residue, later confirmed by the BIAM assay. This complex showed a moderate cytotoxicity (IC_50_ = 3.2 μM on A2780S) and the activity of cellular TrxR was decreased by 2-fold, which is translated into the accumulation of oxidized Trx1 [19]. Among thiolate-substituted NHC gold(I), the heterobimetallic complex **11** displayed a moderate cytotoxic activity (IC_50_ = 10.3 μM on PC-3 cells) associated with apoptotic cell death. TrxR activity in PC-3 cells substantially dropped to 24% after 24 h with the 5 μM complex **11** [99]. Within our list of selected molecules, this complex is the only one to possess antimigratory properties. This feature could be useful to prevent the appearance of metastases [122].

### 4.2. Silver(I) Complexes: NHC^Ag(I)^L

A few NHC silver(I) complexes were also prepared (**13**–**15**). The first complex **13** of our series is the analogue of the gold(I) complex **12** [100]. Both of them showed excellent inhibitory properties on TrxR (respectively, IC_50,TrxR1_ = 5.9 or 12.6 nM; IC_50,TrxR2_ = 19.2 and 67.0 nM) and to a lesser extent on GR (IC_50,GR_ = 65.0 or 85.0 nM). In non-cancerous cells, the two complexes are inactive on TrxR. In contrast to compound **12**, complex **13** is highly active on this enzyme in A2780 cells, with almost complete inhibition reached after 24 h upon incubation with **13** (8 μM). In terms of cytotoxicity on the same cells, the same trend is also noticed (IC_50_ = 3.3 vs. 5.2 μM) [100]. Here again, the preferential localization of **12** and **13** in the nucleus (owing to the anthracene motif) is not in accordance with the observed biological effects. The second complex **14** is an extremely potent and dose-dependent inhibitor of TrxR (IC_50,TrxR_ = 2.4 nM). This compound is only active on A2780 cell lines, especially on A2780 resistant to cisplatin (IC_50_ = 90 nM vs. 0.4 μM for A2780S). This toxicity was related to ROS production. This compound is one of the few examples of molecules for which different mechanisms of action have been searched and identified [102]. This led to the identification of an inhibitory effect of this complex on topoisomerase1, PARP-1. The last complex (**15**) of our list displayed a very low IC_50_ on TrxR1 (IC_50,TrxR1_ = 12.5 nM), but it is also only moderately cytotoxic on MCF-7 cells (IC_50_ = 14.6 μM). This finding could be explained by the presence of the four sulfonate groups that impair cellular uptake [103].

### 4.3. Multi-Modal Au(I) Complexes

Several complexes were designed to combine TrxR inhibition properties to others, i.e., theranostic, redox, targeting or optical properties (like complexes **12** and **13** carrying a fluorescent anthracenyl substituent [103]). As an example, complex **22**, described by Arambula, combines the inhibitory properties of NHC gold complexes to ROS production, allowed by the redox cycling properties of the naphthoquinone moieties. Overall, this complex is highly cytotoxic on A549, A2780 and PC-3 cells (IC_50_ = 73 nM, IC_50_= 26 nM, IC_50_ = 96 nM, respectively) and especially on MDR 2780CP cells (IC_50_ = 54 nM). The inhibition of TrxR in A549 cells was estimated with a value of 2.5 μM after 6 h of incubation [107]. The alkyne^Au^PPh_3_ complex **20** possesses a ligand derived from oleanic acid (pentacyclic terpene), itself endowed with biological properties [105]. This complex is a moderate TrxR inhibitor (IC_50,TrxR_ = 2.7 µM) and has also a moderate toxicity on A2780 cells (IC_50_ = 10 µM). Interestingly, in A2780 cells, this complex exerts its effects via an interaction with mitochondrial TrxR, and also via the induction of endoplasmic reticulum stress (ERS) attributed to ROS production. In cells, TrxR expression is reduced by 55% after 24 h incubation with 15 µM. This complex has been studied in vivo (see Section 4.6). More recently, complex **23** was obtained by grafting a TPE (tetraphenyl ethylene) substituent on the NHC ligand of a gold(I) complex, inspired by Ott’s work [108]. TPE is known to display AIEgenes (aggregation induced emission luminogens) properties [108]. This property enables the ligand to be localized in the cytoplasm and not in the nucleus or mitochondria. Furthermore, 10 µM of complex **23** inhibits by 60% the activity of rat TrxR after 30 min. In addition, its incubation in the presence of siRNAs impairs TrxR expression, resulting in the loss of the complex’s cytotoxicity in cancer cells. This clearly shows that TrxR plays a role in the cytotoxicity of this molecule. The localization of the complex in the cytoplasm suggests that the interaction takes place with cytoplasmic TrxR1 and not with mitochondrial TrxR2. However, this localization seems questionable, since complex **23** causes depolarization of the mitochondrial membrane. The complex also has a radiosensitizer effect, boosting its anti-cancer efficacy to levels superior to those observed with auranofin. Complex **21** is another example of the AIE-active and photosensitive gold(I) complex, including a TBP (N,N-diphenyl-4-(7-(pyridin-4-yl)benzo[c][1,2,5]thiadiazol-4-yl)aniline) entity. It is an extremely potent inhibitor of TrxR (IC_50, TrxR_ = 10 nM). Antitumor activity is boosted by PDT triggered by irradiation [106]. This complex also presents a high in vivo activity (cf. Section 4.6).

### 4.4. Gold(III) Complexes

The Au(III) ion is a harder base than Au(I), and the corresponding complexes, which are isoelectronic of Pt(II) complexes, are expected to be less reactive toward the selenocysteine residue of TrxR. These complexes are well known to be less stable that the Au(I) counterparts because of their proneness to metal reduction and/or ligand exchange [37]. Among all gold(III) complexes, the ones showing a TrxR inhibition property happen to be cyclo-metallated complexes. Historically, complexes **29** and **30** were the first to be reported as good TrxR2 inhibitors (respectively, IC_50,TrxR_ = 0.21 and 0.28 µM) [113,114]. These complexes are fully inactive on mitochondrial GPx and GR [113]. In cells, they display a good cytotoxicity (respectively IC_50_ = 3 and 1 µM on A2780S). Complexes **31** and **32** are even stronger TrxR inhibitors (respectively IC_50,TrxR_ = 18 and 9 nM). In contrast to complex **30**, these complexes are poorly cytotoxic (IC_50_ = 47 µM) because of their low stability in water [115]. Dinuclear cyclometalated gold(III)-phosphine complex **25** is a complex comprising a tridentate C^N^C ligand [109]. It stands out in this series because of its strong inhibition of TrxR (IC_50,TrxR_ = 2.7 nM) and its high cytotoxicity on HepG2 cells (IC_50_ = 0.21 µM). Interestingly, it is much more active on TrxR than its corresponding mononuclear complex **24** (IC_50,TrxR_ = 19 nM). The difference in TrxR inhibition is even more marked in cells, with IC_50,TrxR_ values of 7.3 µM for **25** and over 32 µM for **24**. As downstream effects, the TrxR inhibition elicited ER stress and triggered a death-receptor-dependent apoptotic pathway [109]. Complex **25** has been the subject of in vivo studies (cf. Section 4.6). Complex **27,** comprising a tridentate C^N^C ligand and the same thioglucose ligand as auranofin, displays the same pattern as complex **24.** Unsurprisingly, **27** is a very strong TrxR inhibitor (IC_50,TrxR_ = 3 nM and 60 nM on TrxR1 and TrxR2, respectively), but its antiproliferative effect is low (IC_50_ = 13 µM on SKOV-3) [111]. Another type of cyclometallated Au(III) complex, compound **26**, has a significant cytotoxicity on HeLa cells (IC_50_ = 0.42 µM), but its TrxR inhibitory property is moderate (IC_50,TrxR_ = 1.2 µM), suggesting a limited role of TrxR on its cytotoxicity [110]. For another reason, this is also the case for the bathophenanthroline complex **28**, which elicits a strong antiproliferative effect on A2780 cells (IC_50_ = 0.12 µM) but is a moderate inhibitor of TrxR (IC_50,TrxR_ = 1.2 µM). Indeed, cytotoxicity is, for its most part, attributed to the bathophenanthroline ligand itself (IC_50_ = 0.28 µM), which is promptly released in cells [112].

### 4.5. Piano-Stool Ru, Ir and Os Complexes

A limited number of organometallic complexes with other metal than gold and silver have been reported in the literature. All of them display a half-sandwich (“piano stool”) structure, i.e., they possess a metal-aromatic ring bond (cyclopentadienyl or aryl). The first of this series, the NHC ruthenium complex **33**, prepared by Ott, shows fairly good TrxR inhibition and cytotoxicity (IC_50,TrxR_ = 0.78 µM and IC_50_ = 2.07 µM on MCF-7 cells) [123]. Conversely to gold complexes, complex **33** equally reacts with Cys and Sec [116].

McGowan compared Ru and Ir complexes with ß-ketoiminato (N,O) (**34**, **36**) or naphthoquinone (O,O) (**35**, **37**) ligands [117]. The behavior of these complexes is very similar for both metals. On the other hand, it differs considerably depending on the ligand. TrxR inhibition is more pronounced with (N,O) complexes and is in the nanomolar range for **34** and **36** (IC_50,TrxR_ = 285 and 68 nM, respectively), whereas it is in the micromolar range for (O,O) complexes **35** and **37** (IC_50,TrxR_ = 1.4 µM). Cytotoxic effects follow the same trend, with the toxicity of (N,O) complexes below 5 µM but above 18 µM for (O,O) complexes, although this effect alone cannot explain the cytotoxicity of the complexes. Indeed, studies of the complexes’ effect on DNA show that **34** and **36** induce DNA SSBs (Single-Strand DNA Breaks). These complexes therefore appear to act via an original mechanism of action, different from that of cisplatin, which is known to induce DSBs (Double-Strand DNA Breaks). The pyrithione ruthenium complex **38** shows a significant toxicity on three different A2780 subtypes (IC_50_ = 1.0–1.6 µM) and reduces motility. It does not induce ROS production nor mitochondrial impairment, and only a low TrxR inhibition (IC_50,TrxR_ = 4.1 µM). Thus, the involvement of the TrxR pathway is likely quite limited in the mechanism of action of this compound [118]. The last complex of the series, the osmium complex **39,** also shows a significant cytotoxicity on MDA-MB-231 cells (IC_50_ = 2.4 µM) and a decrease of about 30% of TrxR activity in cells (5 µM, 6 h), consistently with its ability to form a covalent adduct with cysteine [119].

### 4.6. In Vivo Studies

Quite surprisingly, among the 39 selected complexes, only 6 of them (**9**, **16**, **19**, **20**, **21** and **25**) were studied in vivo (Table 4), likely because of a tendency of these complexes to irreversibly bind to serum albumin (by ligand exchange with its available cysteine residue), decreasing their bioavailability, and their lipophilic character is poorly compatible with intravenous (*iv*) administration. One could assume that adequate formulation in nanoparticles could favor their translation to in vivo testing by improving drug delivery to tumors [39,124].

Among the numerous NHC–gold(I) complexes prepared by Ott, only complex **16** was selected for in vivo studies. The complex was directly injected into the tumor but no tumor regression was observed [95]. Better results were obtained with the heteroleptic (Bis)NHC gold complex **9** on Ishikawa cells (endometrial cancer cells) xenografts. *Ip* administration of **9** induced a larger decrease in tumor growth compared to auranofin (IRT = 20% at 10 mg/kg). Interestingly, the analysis of the tumors harvested from mice treated with **9** showed a decreased expression of TrxR and Nrf2, while the level of Nrf2 was unchanged for auranofin [98]. A decrease in TrxR expression in tumors was also observed upon *ip* administration of complex **20**, which also yielded a significant inhibition of tumor growth [105]. Regarding compound **21**, in vivo studies were carried out on HeLa cell xenografts and *intratumoral* injections. Strong tumor regression was observed, which was even stronger when combined with irradiation with white light owing to the PDT properties of **21** [106]. Complex **19** was administered by *iv* to nude mice carrying MCF-7 xenografts. With an IRT of 91.5%, complex **19** appears to show the highest antitumoral activity of the series [104].

## 5. Ferrocifens as TrxR Inhibitors

Ferrocifens are a family of ferrocenyl derivatives developed by Jaouen’s group that derive from hydroxy-tamoxifen (OH-Tam), **40**, the active metabolite of tamoxifen, a molecule widely used in the treatment of hormone-dependent breast cancers ([Fig cancers-15-04448-ch003]) [125]. The initial idea was to replace the ß aromatic ring of tamoxifen with a ferrocenyl unit and to study the biological changes associated with this substitution.

The very first compound of the series, **43**, the ferrocenyl derivative of **40**, had a dual effect on MCF-7 (hormone-dependent breast cancer cells), being both anti-estrogenic as hydroxy-tamoxifen but also antiproliferative, a property not observed for **40** [126]. Many complexes have since been synthesized and their antiproliferative activity has been initially studied in MCF-7 and MDA-MB-231 (hormone-independent) breast cancer cells, the latter one being representative of TNBC, an MDR cancer cell line. Later on, studies were extended to other cell lines, including glioblastoma from rat [127] or more recently to human glioblastoma PDCLs (patient derived cell lines) [9]. These results have been the subject of several reviews [128,129,130]. Soon after the discovery of these interesting properties, research was undertaken to unravel the mechanism of action of these complexes. This research highlighted the essential role played by the unique redox properties of ferrocene and by the ferrocene-*p*-ene-phenol motif. Indeed, the starting point for the antiproliferative activity of ferrocifens seems to be the oxidation in cells of Fe(II) to Fe(III) followed by the abstraction of a proton after a new oxidation and the abstraction of a second proton to a quinone methide (Figure 7).

Quinone methides are electrophilic entities that can undergo Michael additions with nucleophiles, like the selenocysteine of TrxR’s active site [128]. Michael addition to quinone methides resulting from the oxidation of organic flavonoids has been also hypothesized by Holmgren to explain their TrxR-inhibiting properties [131]. In the case of ferrocifens, it was shown that the quinone methide prepared by the chemical oxidation of **41** could undergo a 1,8-Michael addition with β-mercaptoethanol to yield **50,** whose structure was confirmed by X-ray diffraction analysis (Figure 8) [132].

According to this mechanism, ferrocifens appear as pro-drugs, which are converted within cells into quinones methides, thanks to the particular redox properties of ferrocene. These quinones methides are reactive molecules; therefore, they are not stable and, among ferrocifens, only the quinones methides (QMs) of **41** and **43** have sufficient stability to be isolated in the solid state after chemical oxidation with Ag_2_O (Figure 8) [133]. In contrast, chemical oxidation of the ansa-complex **44** generated a product too unstable to be isolated [134]. The first in vitro results of TrxR inhibition by ferrocifen-QMs were obtained with these two stable QMs (**41-QM**, **43-QM**) and compared to the values obtained for the starting complexes [133] (Figure 9).

As shown in Figure 9a, **41** and **43** are only weak inhibitors of TrxR1 (IC_50,TrxR1_ around 15 µM), while their corresponding QMs are much stronger TrxR1 inhibitors (IC_50,TrxR1_ = 2.6 and 2.2 µM, respectively) [133]. Alternatively, the quinone methides of the three ferrocenyl complexes **42**, **43**, **45** ([Fig cancers-15-04448-ch004]) were generated in situ by enzymatic oxidation in the presence of a mixture of HRP (horseradish peroxidase) and H_2_O_2_ [135]. Enzymatic oxidation of the ansa-complex **44** afforded the quinone methide radical, which is considered as the active species on TrxR [136]. The enzymatic oxidation products of the osmium complexes **45** and **46** are quinone methide carbocations ([Fig cancers-15-04448-ch004]) [137]. The IC_50_ values of TrxR1 inhibition in vitro, obtained for the complexes and their corresponding QMs generated in situ by the HRP + H_2_O_2_ mixture, are summarized in Table 5.

These results confirm that only the QMs generated by enzymatic oxidation are effective inhibitors of TrxR. The most efficient QMs (**42**, **43**, **45**) have IC_50_ values in the nanomolar range (30–60 nM), i.e., comparable to the most efficient gold complexes (cf. Table 1 and Table 2). Cationic osmium-derived QMs (**46-QM^+^**, **47-QM^+^**) are less potent inhibitors than ferrocenyl QMs (IC_50,TrxR_ = 2.4 and 1.2 µM). BIAM studies show that all these complexes, except for **43,** are covalently linked to both the selenocysteine and the accessible cysteine residues of TrxR. These complexes do not inhibit GR nor GPx, likely because its selenocysteine residue is less accessible [133].

In this family of metallocifens, a negative correlation is observed between IC_50_ values on TrxR and cytotoxicity on MDA-MB-231 cells (Table 5). This is, for example, the case for **44**, the ansa-complex, which is the most cytotoxic of the series (IC_50_ = 0.089 µM), but only a moderate inhibitor of TrxR (IC_50,TrxR_ = 0.15 µM). On the contrary, **45** is a strong inhibitor of TrxR (IC_50,TrxR_ = 30 nM), while its cytotoxicity on cancer cells is quite modest (IC_50_ = 6 µM). The high cytotoxicity of the ansa complex (**44**) seems to be associated with the peculiar lifetime of its QM-radical [136]. Regarding **45**, its lower cytotoxicity has been attributed to the fact that it can only adopt a *cis* configuration, while the active configuration of the ferrocene-*p*-ene-phenol motif is associated with the *trans* form [15].

The quinone methides derived from the two osmium complexes, **46** and **47**, have low IC_50_ values on TrxR, but only the tamoxifen-like complex **47** has a significant cytotoxicity on cells (Table 5). These differences in cytotoxicity cannot be related to their lipophilicity values, which are in the same range for these different complexes [139], but could be attributed to the fact that, at physiological pH, the amino-terminated side chain of the Tam-like complexes is protonated, giving them a lipophilic cation status. In addition, depolarization of the mitochondrial membrane (MMP) has been observed only for these Tam-like metallocifens, a stronger effect being observed with the iron with respect to the osmium complex [138]. As described for the NHC–gold(I) complexes, these molecules are localized in the right place to interact with mitochondrial TrxR, leading to cell death. This hypothesis is supported by the fact that the quantification of iron or osmium by ICP-OES in cells incubated with the Tam-like complexes, **43** and **47**, showed their preferential accumulation in the crude nuclear fraction (around 50%) and mitochondria (40%), but not in the cytosol (less than 10%) [138]. The incubation of **42**, **43**, and **44** in cells generates a fast production of ROS (10 min incubation, in MDA-MB-231) measured with the fluorescent probe H_2_DCFDA, while ferrocene alone and hydroxy-tamoxifen **40** do not generate ROS [130]. However, there is no correlation between the level of ROS found after incubation of complexes in cells and their cytotoxicity. The best example is provided by **48**, the ferrocifen lacking phenol substituents. Complex **48** produces one of the highest levels of ROS in a selection of potent ferrocifens (**42–44**), while its cytotoxicity is average (IC_50_ = 7.5 µM) [130]. Taken together, all these observations seem to validate the assumption that these complexes act via different mechanisms involving several targets [130].

The activity of TrxR in Jurkat cells incubated with **41**, **43** and their corresponding QMs has been measured [133] (Figure 9b). Both **43** (Fc-OH-Tam) and its quinone methide are quite efficient TrxR inhibitors (around 55% inhibition in the presence of 20 µM of complexes). This result clearly demonstrates that TrxR plays a role in the cytotoxicity of **43** in cells. In addition, this experiment indirectly confirms that, in the cells, **43** is transformed into its quinone methide, as they both inhibit TrxR in the same way. Let us also notice that **43** inhibits TrxR2 (mitochondrial form) more than TrxR1 (cytosolic form) (72% and 35%, respectively, in the presence of 30 µM of the complex) [138]. This result is opposite to the in vitro results for **43-QM** prepared by chemical oxidation [133], but consistent with the cell distribution of the complex mentioned above [138].

In contrast, neither **41** (Fc-mono-OH) nor its quinone methide **41-QM** inhibit TrxR in cells (Figure 9b). Thus, TrxR does not seem to play a role in the cytotoxicity of **41** in Jurkat cells. This has been related to the fact that, in cells, the quinone methide of **41** is readily transformed into its indenes **51** and **52** (Figure 10), a molecule that is unable to undergo Michael addition and consequently cannot interact with the selenocysteine of TrxR [97]. On the other hand, indene has been shown to be an inhibitor of cathepsin B, a cysteine protease [140]. Cathepsin B could therefore be another target of ferrocifens.

This study of TrxR inhibition evidenced that, conversely to what was initially suggested [8], Tam-like and phenolic complexes probably do not act through the same mechanisms of action. Indeed, the DLC character of Tam-like complexes seems to be the key factor for efficient TrxR inhibition in cells. This inhibition, which plays an important role for Tam-like complexes, does not apply for phenolic complexes. In any case, it seems clear that these complexes exert their cytotoxicity via several targets [128,130].

As mentioned above (cf. Section 4.6), lipophilic molecules are not suitable for direct in vivo administration. Regarding ferrocifens, this problem was overcome thanks to their formulation in lipid nanocapsules (LNCs) [127]. These LNCs consist of a lipid core (oil or triglycerides) in which the active principle is soluble, surrounded by a lipid surfactant (Lipoid) and one or two layers of PEG (functionalized polyethylene glycol), which provides the interface between the lipid phase and the aqueous phase. The LNCs obtained can be injected into physiological saline. In addition, the judicious choice of PEG makes it possible to obtain stealth LNCs, which are not detected by macrophages and thus remained longer in the bloodstream [141,142]. Complex **42**, formulated in LNCs, has been used in several in vivo studies performed in rats, on 9L rat glioblastoma tumors both ectopic or orthotopic [127,142,143]. An excellent IRT value (98%) was for example obtained on ectopic tumors after the I.V. injection of stealth LNCs (2.4 mg per rat, every other day, ×10) [142]. On an orthotopic model, coupling chemotherapy with **42** formulated in LNCs (0.36 mg/rat, 1 injection by CED in the tumor) and radiotherapy (external irradiation 3 × 4 Gy) increased the medium survival time of animals from 25 to 40 days with two long-term survival (animals still alive after 100 days) [143]. Concerning the Tam-like complex **43**, an in vivo experiment was performed on MDA-MB-231 xenografted tumors in nude mice treated with *ip* injections of **43** formulated in LNCs (20 mg/kg) [141]. In this case, a significant IRT value of 36% was observed. All these results show that the formulation of ferrocifens in nanocapsules is well suited to obtain good in vivo effects with these lipophilic molecules.

## 6. Comparison of the Mechanism of Action of Gold Organometallic Complexes and Metallocifens

TrxR is one of the key players in the maintenance of cellular redox balance. Owing to their peculiar metabolism, cancer cells naturally produce a higher level of ROS than normal cells. Consequently, TrxR is generally upregulated, which makes it a meaningful target to design selective anticancer drugs.

At the molecular level, TrxR displays a unique, solvent-accessible, C-terminal redox active site comprising a rare selenocysteine known to be highly nucleophilic. As a consequence, most of the reported TrxR inhibitors have an electrophilic character and form covalent adducts with the enzyme [144]. Transition metal-based compounds form a wide class of TrxR inhibitors, among which gold complexes are the most abundantly represented, and give rise to IC_50,TrxR_ in the nanomolar range. From a purely chemical point of view, the potent inhibiting property of gold and silver complexes is explained by the HSAB (Hard and Soft Lewis Acids and Bases) principle that defines Au and Ag as soft acids and S and Se as soft bases.

For NHC-gold(I) complexes **3**–**5**, the IC_50,TrxR_ is directly related to the lability of the second ligand. Complexes **4** and **5** display the same level of cytotoxicity, but **4** is significantly less active on TrxR than **5**, illustrating that other parameters need to be taken into account to rationalize the anticancer properties (stability and lipophilicity to name a few). Except for **10**, **12** and **13**, the most effective compounds are selective to TrxR, having much lower or even no effect on the structurally related enzyme GR lacking the C-terminal redox active site. When tested, none of the complexes inhibit the selenoenzyme GPx, indicating that the Cys-Sec dyad is crucial for inhibition. Indeed, linear Au(I) (and probably Ag(I)) complexes were hypothesized early on to undergo a sequential ligand substitution to afford a Cys–Au–Sec adduct, abolishing the reductase property of TrxR. Mass spectrometry analysis of mixtures of the C-terminal dodecapeptide sequence of TrxR and the NHC-gold(I) complexes **3**–**5** confirmed the formation of several adducts by the displacement of one or both ligands only for complexes **3** and **4** [145]. The reaction of the gold(III) complex **30** with the same dodecapeptide was also investigated by MS, and it was found that this complex underwent reduction of Au(III) to Au(I) and subsequent decoordination of the tridentate ligand to afford the gold-peptide adduct, likely by the coordination of S and Se [146].

The molecular mechanism of TrxR inhibition by metallocifens dramatically differs in that it involves the organometallic entity only in an indirect way. For the inhibition to occur, metallocifens first need to be oxidized into electrophilic quinone methides, which is made possible owing to the presence of the redox active metallocenyl substituent. In cell-free conditions, all the oxidation products of metallocifens are potent inhibitors of TrxR with IC_50_ in the low nanomolar range. Inhibition takes place via the Michael addition of the selenol and/or the thiol of the active site Sec and Cys. This mechanism strongly resembles that of the flavonols quercetin and myricetin that readily oxidize into semiquinone or even QM, providing a plausible explanation for their antiproliferative activity [131]. In cells, TrxR inhibition only takes place with **43** (Fc-OH-Tam) and not with the mono-phenol or the diphenol derivatives, stressing the importance of the dimethylamino-terminated side chain protonated at physiological pH, which confers a DLC character to the complex, a common feature with some of the gold complexes.

At the cellular level, mitochondrial function impairment (mitochondrial respiration blockage, mitochondrial membrane depolarization) is often highlighted, mostly associated with the DLC character of the complexes that favors accumulation in the mitochondria. This trait is common to some of the gold complexes and the ferrocifen **43**. Regarding ROS production, it has been reported, for Au(I) complexes, as the consequence of TrxR inactivation, while, for ferrocifens, ROS production seems rather associated with the first step of activation, namely, the oxidation of Fe(II) to Fe(III) (Figure 7).

## 7. Discussion and Conclusions

In order to give a visual, synthetic view of the results given in Table 1, Table 2, Table 3 and Table 5, a scatter plot was generated by plotting IC_50_ values on TrxR, measured in vitro, versus IC_50_ values on cancer cells (Figure 11).

Even if the complexes have not always been tested on the same cell lines, this graph clearly evidences a strong heterogeneity among the complexes covering a wide range of IC_50_ on TrxR (2.8 nM to 4.9 µM) and on cancer cells (26 nM to 47 µM). Nevertheless, five groups gathering complexes with similar behavior can be identified. The first group (group 1) includes three chemically unrelated complexes (**14**, **19**, **25**) showing a correlation between high TrxR inhibition (IC_50,TrxR_ in the range 2.4–2.8 nM) and high cytotoxicity on cancer cells (IC_50_ in the range 0.03–0.44 µM). For these complexes, TrxR inhibition can be considered to play a major role in their cytotoxicity. On the contrary, group 2 gathers four complexes (**20**, **35**, **37**, **46**) displaying both low TrxR inhibition and cytotoxicity. Group 3 includes four complexes (**15**, **24**, **27**, **31**, **32**) showing a high TrxR inhibition (IC_50,TrxR_ in the range 3–19 nM) but a poor cytotoxicity on cancer cells (in the range 13–47 µM). For these complexes, their strong inhibition of TrxR observed in vitro is therefore not translated into a high cytotoxicity in cells. This result evidences that IC_50,TrxR_ measured in vitro is not a good predictor of cytotoxicity of the complexes. Indeed, other factors such as good stability in culture medium and in cells, adequate lipophilicity allowing cell uptake, and metabolism also play a role. Group 4 includes three complexes (**8**, **22**, **28**) showing high cytotoxicity (in the range 0.026–0.12 µM) but limited inhibition of TrxR (in the range 0.7–2.5 µM). For these complexes, TrxR is probably not a key player in their cytotoxicity. Group 5 consists of four complexes (**16, 18, 42, 43**) that behaves similarly to auranofin. Of note, metallocifens appear distributed very randomly among the other complexes, two of them belonging to the “auranofin-like” group. Thus, even if they behave differently at the molecular level, they do not form a distinct group in this set of organometallic molecules.

In conclusion, the driving force of our selection of complexes was to pick up complexes representing the state of the art in the domain, leading to a large array of organometallic complexes with IC_50_ values on TrxR and on cancer cells ranging from the nanomolar to micromolar range (Figure 11). The involvement of TrxR inhibition in the expression of cytotoxicity has been unambiguously demonstrated in a limited number of chemically unrelated complexes (group 1). For all the other complexes, no correlation could be drawn. Therefore, it seems reasonable to consider that most of these complexes exert their cytotoxicity through multiple biological targets. Therefore, most of these complexes meet the current trend aiming at a multi-targeted therapeutic approach, and should be suitable candidates to tackle MDR tumors.

## Data Availability

The data presented in this study are available in the references.

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
