# Peer review of "Thioredoxin Reductase and Organometallic Complexes: A Pivotal System to Tackle Multidrug Resistant Tumors?"

_cancers, 2023, doi:10.3390/cancers15184448_

Round 1

Reviewer 1 Report

The reviewed work is very well written. The approach to the topic proposed by the authors is original and brings important insights that can be used by researchers in their work on new drugs. Below are some minor comments of a rather editorial nature. I recommend the review for publication as revised.

Details comments

Figure 3 – please magnified figures

p.5 line 195 – units for IC50 are missing

p.5 line 203, please put capital R in Trxr

p. 6 line 220 FITExP – this abbreviation needs to be better explained

Figure 5 – the resolution is poor

p.7 line 273 after “gold” put the word “ion”

p. 12 line 391 the sentence: “This compound is a rare example of a molecule showing multiple mechanisms of action” – are you sure that other compounds do not show such behavior or maybe there were not tested on other targets – please reconsider this sentence

Author Response

Dear Editor,

First of all, we would like to thank the reviewers for their kind comments on our paper.

According to their comment we have made the following modifications. As requested, they are highlighted in yellow in the corrected version 

Answers to reviewer 1

Figure 3 : labels for the selected amino acids have been enlarged.

Line 195  : µM, the unit for IC50, has been added 

Line 203 : Trxr has been changed for TrxR

Line 220 the abbreviation FITExP has been explained. It means:  Functional Identification of Target by Expression Proteomics

Figure 5 : resolution of the figure has been increased.

Line 273 : ion has been added

Line 392 : the sentence has been rewritten

Reviewer 2 Report

The manuscript by Vessières and coworkers recall the main results obtained with the thioredoxin reductase (TrxR) inhibitor Auranofin of as a strategy to treat the MDR tumors. To this end, the authors focused on gold(I) and gold(III) complexes and metallocifens, including organometallic complexes of Fe and Os derived from tamoxifen. The authors discussed on molecular mechanisms and relationships between TrxR inhibition and cytotoxicity.

I have found this review manuscript well organized. The chapters have been enriched with pictures and chemical drawing, leading the reader trough easier understanding of each discussed subject. The statements have been well documented by literature.

This manuscript can be considered for publication in this Journal without additional changes.

Good, minor spelling corrections

Author Response

Reviewer 2 does not suggested any modification. 
